# POLICY DISENTANGLED VARIATIONAL AUTOENCODER

## ABSTRACT

Deep generative models for video primarily treat videos as visual representations of agents (e.g., people or objects) performing actions, often overlooking the underlying intentions driving those actions. In reinforcement learning, the policy determines actions based on the current context and is analogous to the underlying intention guiding those actions. Through the acquisition of policy representations, we can generate a video capturing how an agent would behave when following a specific policy in a given context. In this paper, we aim to learn the representation of the policy without supervision and the dynamics of the environment conditioned to the policy. We propose *Policy Disentangled Variational Autoencoder* (PDVAE) which can generate diverse videos aligned with the specified policy where the user can alter the policy during the generation. We demonstrate PDVAE with three video datasets: Moving MNIST, KTH action dataset, and VizDoom.

## 1 INTRODUCTION

Videos consist of temporally consistent images and exhibit diverse temporal variations in their visual signals, resulting in numerous semantic features. Deep generative models have effectively captured these semantic features in latent representations for video generation, with motion and content being common approaches for the representation learning (Tulyakov et al., 2018; Wang et al., 2020; 2021; Hyun et al., 2021; Khan & Storkey, 2022; Skorokhodov et al., 2022). The motion representation captures the dynamic changes within the video whereas the content representation encodes the static visual information. Some methods have focused on learning the representation of actions between consecutive frames of video in discrete space to control the generation process (Kim et al., 2020; 2021; Menapace et al., 2021). However, these approaches often overlook the intention behind the actions performed by objects or individuals in the video, viewing them primarily as visual signals.

Distinguishing between different intentions or behavioral goals behind the same action is challenging. For instance, on Figure 1, these players may perform identical actions initially, but their subsequent actions diverge based on their behavioral goals. Distinguishing intentions or behavioral goals requires a deep understanding of context, the agent's decision-making, and the environment. The action representations, which focus solely on frame-to-frame action, are inadequate for distinguishing the behavioral goal of a single action. Similarly, motion representations that emphasize spatiotemporally consistent motion trajectories struggle to differentiate between distinct trajectories by the intention behind the actions. To address this issue, different types of representations that can account for a player's specific behavioral goals are needed. By learning such representation, we can generate a video of a player converting from time attacker to treasure hunter in front of the entrance to a boss.

The intention behind an action is analogous to the policy in reinforcement learning, representing an agent's decision-making process in various situations. This policy can be seen as the cognitive process guiding an agent's actions, while the video serves as the observations of an agent (e.g., a person or object). In this paper, we introduce a model that learns the representation of policy without labels and the dynamics of the environment characterized by the policy. The model can distinguish the video by the policy of an agent and generates a video aligned with the policy. For instance, the model can differentiate between gameplay videos where a player's objective is to either achieve a high score or finish a session quickly. The model produces diverse videos where each agent demonstrates distinct actions that adhere to the specified policy. Furthermore, by altering the policy from one to another, the model has the capacity to produce a video that is not present in the sample dataset.

The concept of a generative model that takes into account the environment's dynamics is not new to the literature. Temporal-difference variational autoencoder (TD-VAE) (Gregor et al., 2019) is

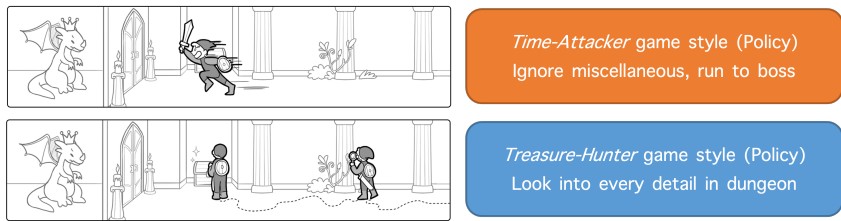

Figure 1: **Different game styles (policies) by players**. **Upper**: *time-attacker* ignores the miscellaneous objects in the dungeon, and run straight to the objective of the game. **Lower**: *treasure-hunter* enjoys the easter eggs planted by developer and tries to find out every treasure.

designed to learn the representation of the dynamics of the environment through a state-space model. TD-VAE encodes input video frames into a *state* that encompasses all relevant information about the environment for the prediction. The policy-related information is also embedded in the state, hence the model cannot distinguish the underlying behavioral goal of an agent.

We propose *Policy Disentangled Variational Autoencoders* (PDVAE) which extracts the policy of an agent in a video and generates video conditioned to the policy and the state encoded from the past frames. We assume the videos are the observations of agents, in which each agent acts upon own behavioral goal. PDVAE learns the goal as a policy and categorizes it into discrete numbers. We add the term, policy, in the derivation of the ELBO of TD-VAE to acquire the ELBO of our model. As the ELBO of PDVAE indicates, we use auxiliary modules to extract the policy from the video and use it as an additional input to TD-VAE. We have added a regularization term to prevent the state from having information on the policy, achieving the disentanglement of the state and the policy. The generated video of PDVAE maintains the background and the figure of the agent while the agent acts according to the given policy. Through qualitative and quantitative experimental validations, we verify the effectiveness of PDVAE.

## 2 PRELIMINARIES

A partially observable Markov decision process (POMDP) (Smallwood & Sondik, 1973; Kaelbling et al., 1998) is a common reinforcement learning framework for modeling agents that interact in partially observed environments. Formally, it is defined as a tuple $\langle Z, A, X, T, O, R \rangle$ where $Z$ is the state space, $A$ is the action space, $X$ is the observation space, $T$ is the transition probability, $O$ is the observation probability, and $R$ is the reward function [1]. The transition probability $T$ describes the dynamics of the environment by mapping the state and action to another state, and the reward function $R$ represents the agent's behavior goal to act. The state contains the full information about the world or environment but cannot be known, so the agent maintains the probability distribution, called belief $b_t$, over the states with the past observation history $x_{<t}$ or $x_{1:t}$ at time $t$. Policy $\pi$ represents the agent's behavioral rule as a function of given the belief $b_t$ to the action $a_t$.

Temporal-difference variational autoencoder (TD-VAE) (Gregor et al., 2019) is a sequence generative model which assumes the POMDP framework to generate future observation sequence. TD-VAE derives the evidence lower bound (ELBO) from the conditional likelihood $p(x_t|x_{<t})$ of an observation $x_t$ at time $t$ given the past observations $x_{<t}$, by inferring over two states $z_{t-1}$ and $z_t$ as follows:

$$\log p(x_t|x_{<t}) \geq \mathop{\mathbb{E}}_{(z_{t-1},z_t) \sim q(z_{t-1},z_t|x_{\leq t})} \Big[ \log p(x_t|z_t) + \log p(z_{t-1}|x_{<t}) - \log q(z_t|x_{\leq t}) \quad (1)$$
$$+ \log p(z_t|z_{t-1}) - \log q(z_{t-1}|z_t, x_{\leq t}) \Big].$$

From the above belief-based ELBO in Equation 1, the following loss can be obtained:

$$\mathcal{L}^{\text{TD-VAE}} = \mathop{\mathbb{E}}_{\substack{z_t \sim p_B(z_t|b_t) \\ z_{t-1} \sim q_S(z_{t-1}|z_t, b_t, b_{t-1})}} \Big[ \log p_D(x_t|z_t) + \log p_B(z_{t-1}|b_{t-1}) - \log p_B(z_t|b_t) \quad (2)$$
$$+ \log p_T(z_t|z_{t-1}) - \log q_S(z_{t-1}|z_t, b_t, b_{t-1}) \Big].$$

---

[1] In reinforcement learning, it is a common practice to represent the state and observation space as $S$ and $O$, respectively. In our generative model framework, we adopt the notation $Z$ and $X$ to refer to these spaces.

In the above two equations, $p(x_t|z_t)$ in Equation 1 corresponds to the probability of observing $x_t$ given the latent state $z_t$, which can be interpreted as the *decoder network* $p_D$ in Equation 2 from the generative model perspective. Since the past observations $x_{<t}$ and $x_{\leq t}$ become the belief states $b_{t-1}$ and $b_t$ respectively, despite the difference between $p$ and $q$, both $p(z_{t-1}|x_{<t})$ and $q(z_t|x_{\leq t})$ in Equation 1 can be regarded as one *belief network* $p_B$ as $p_B(z_{t-1}|b_{t-1})$ and $p_B(z_t|b_t)$ in Equation 2. The filtering distribution $p(z_t|z_{t-1})$ and the smoothing distribution $q(z_{t-1}|z_t, x_{\leq t})$ in Equation 1 can be denoted as the *transition network* $p_T(z_t|z_{t-1})$ and *inference network* $q_S(z_{t-1}|z_t, b_t, b_{t-1})$ in Equation 2 by using the filtering and smoothing technique.

TD-VAE employs a common network architecture, $D$ block, to output the parameters for the distributions $p_B, q_S$, and $p_T$ [2]. Each network takes the conditioned variables as an input and outputs the mean and log standard deviation of the normal distribution. For the belief network $p_B(z_t|b_t)$, the $D$ block utilizes the encoded $b_t$ obtained from the forward RNN as input, generating a normal distribution from which the state $z_t$ is sampled using reparameterization trick (Kingma & Welling, 2013). The transition network $p_T$ and the inference network $q_S$ utilize a similar $D$ block architecture, yet their weights are not shared. Both networks produce a normal distribution of state as their output.

## 3 POLICY DISENTANGLED VARIATIONAL AUTOENCODER (PDVAE)

Since the transition network of TD-VAE solely uses the state for the generation of future sequences, we interpret that the state encapsulates all relevant information for the prediction, including the policy. The aim of this paper is to generate videos aligned with the specified policy, whether the policy of input video and the generation is the same or not. We propose *Policy Disentangled Variational Autoencoder (PDVAE)* which generates such video by learning the disentangled representation of the policy and the state. In subsection 3.1, we add policy $\pi$ to derive ELBO, which is the theoretical basis of PDVAE. In subsection 3.2, we describe the policy extraction module that can unsupervisedly extract the agent's policy without label information about which policy the collected sequence data was generated from. In subsection 3.3, we proposes a method for integrating policies in the form of extracted code vectors into TD-VAE. Finally, subsection 3.4 presents the PDVAE's overall training and generation procedures.

### 3.1 DERIVATION OF ELBO

We extend TD-VAE to PDVAE by incorporating the policy $\pi$ into the Equation 1. The Equation 1 uses the posterior distribution $q(z_{t-1}, z_t|x_{\leq t})$ to derive the ELBO from the conditional likelihood of observations given the past observations $x_{\leq t}$. In the case of PDVAE, we modify this by adding the policy $\pi$ to the posterior distribution, resulting in $q(z_{t-1}, z_t, \pi|x_{\leq t})$, as illustrated in the following Equation 3 [3]:

$$\log p(x_t|x_{<t}) \geq \mathop{\mathbb{E}}_{\substack{(z_{t-1}, z_t, \pi) \sim \\ q(z_{t-1}, z_t, \pi|x_{\leq t})}} \Big[ \log p(x_t|z_t) + \log p(z_{t-1}|x_{<t}) - \log q(z_t|x_{\leq t}) \tag{3}$$

$$+ \log p(z_t|z_{t-1}, \pi) - \log q(z_{t-1}|z_t, \pi, x_{\leq t}) + \log p(\pi|x_{<t}) - \log q(\pi|x_{\leq t}) \Big].$$

By following the same process as in Equation 1 and 2, we can obtain Equation 4 from Equation 3.

$$\mathcal{L}^{\text{PDVAE}} = \mathop{\mathbb{E}}_{\substack{z_t \sim p_B(z_t|b_t) \\ z_{t-1} \sim q_S(z_{t-1}|z_t, \pi, b_t, b_{t-1})}} \Big[ \log p_D(x_t|z_t) + \log p_B(z_{t-1}|b_{t-1}) - \log p_B(z_t|b_t) \tag{4}$$

$$+ \log p_T(z_t|z_{t-1}, \pi) - \log q_S(z_{t-1}|z_t, \pi, b_t, b_{t-1}) \Big]$$

$$+ \mathop{\mathbb{E}}_{\pi \sim q_P(\pi|x_{\leq t})} \Big[ \log p(\pi|x_{<t}) - \log q_P(\pi|x_{\leq t}) \Big]$$

Remarkably, we note two important facts. Firstly, in the first expectation, we observe that the policy $\pi$ is explicitly separated within the environment's dynamics $p_T$ and $q_S$. Furthermore, the first

---

[2]Detailed network architecture for $D$ block is in the appendix.
[3]Detailed derivation for the ELBO is in the appendix.

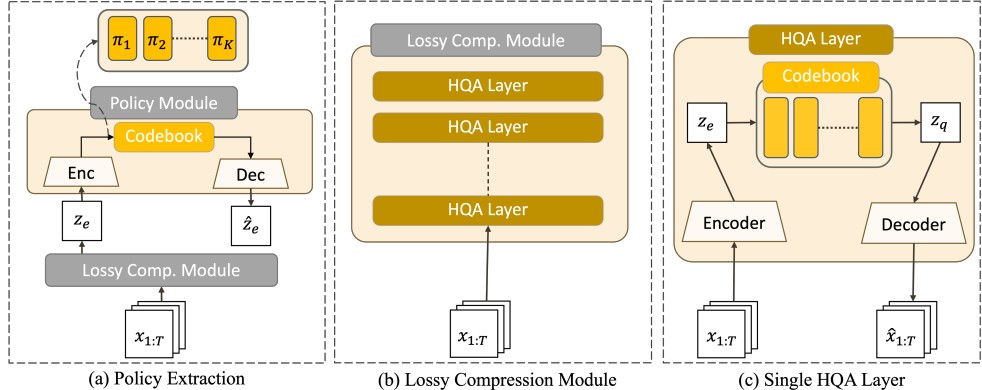

Figure 2: **(a)** the modules for policy extraction from the video, where the policy module is the single HQA layer with the codebook of size $K$. **(b)** the lossy compression module which consists of stacked HQA layers. **(c)** the single HQA layer used in both modules.

expectation takes the policy and the observations as the input whereas the second expectation solely learns the policy from the observations. As the Equation 4 suggests, we use an auxiliary module to separately learn the representation of policy and add the policy in TD-VAE.

## 3.2 EXTRACT POLICY FROM VIDEOS

We extract the policy from the video with two modules detailed in Figure 2: a lossy compression module and a policy module. We consider the videos capture an agent acting upon policies $\pi_i, i \in \{1, ..., K\}$, where the number of policy $K$ is a hyper-parameter specified before the training. Both modules utilize Hierarchical Quantized Autoencoders (HQA), a model designed for extreme lossy compression (Williams et al., 2020).

HQA consists of multiple layers, each including a CONV2D encoder, decoder, and codebook layer for quantization. The model is trained in a greedy layer-by-layer fashion, where each layer begins training after the preceding layer has completed its training. Each layer of HQA depicted in Figure 2(c) is trained with the following loss where the first term is the *reconstruction loss*, the second is the *entropy* of $q$, and the third is *probablistic commitment loss*:

$$\mathcal{L}^{\mathrm{HQA}} = -\log p(x|z = k) - \mathcal{H}[q(z|x)] + \mathbb{E}\,||z_e(x) - e_z||_2^2. \tag{5}$$

The lossy compression module illustrated in Figure 2(b) preserves the global semantic features, including policy, and loses the local features. To preserve the features related to the temporal dimension, we use the timestep of the video as the channel dimension $x_{1:T} \in \mathbb{R}^{H \times W \times T}$, hence we convert the video to the greyscale. Each layer of the module compresses the video by a factor of two in terms of $H$ and $W$. The module is stacked until it encodes the observations to $z_e \in \mathbb{R}^{2 \times 2 \times d_\pi}$ where $d_\pi$ is the dimension of the code vector. The lossy compression module is pre-trained before the training of PDVAE. The module encodes the video to $z_e$ which is the input for the policy module.

The policy module learns the categorical distribution of policy $p(\pi_i|x_{<t})$ and is jointly trained with the TD-VAE built upon the first expectation term in Equation 4. The second expectation term is replaced with the Equation 5 for the training. The policy module extracts the policy from the video with the constraint on the codebook and the usage of policy in the TD-VAE. We set the number of code vectors in the codebook of the policy module to $K$, where the code vector serves as the policy. The policy module encodes $z_e$ to $z_p \in \mathbb{R}^{1 \times 1 \times d_\pi}$ and quantizes to a code vector, which serves as the policy. The constraint alone is not sufficient for extracting the policy from the observations. We use modified TD-VAE to model the dynamics of the environment conditioned to the policy. The modified TD-VAE serves as a regularizer for the policy module to preserve policy-related information in the code vector.

## 3.3 ENVIRONMENT DYNAMICS CONDITIONED TO POLICY

PDVAE employs a modified version of TD-VAE to learn environmental dynamics, featuring two key alterations. Firstly, the transition and inference networks now take policy $\pi$ from the policy module as an additional input as illustrated on Figure 3. The difference between the first expectation term of PDVAE loss (Equation 4) and TD-VAE loss (Equation 2) is the conditioned policy $\pi$ in $p_T$ and $q_S$.

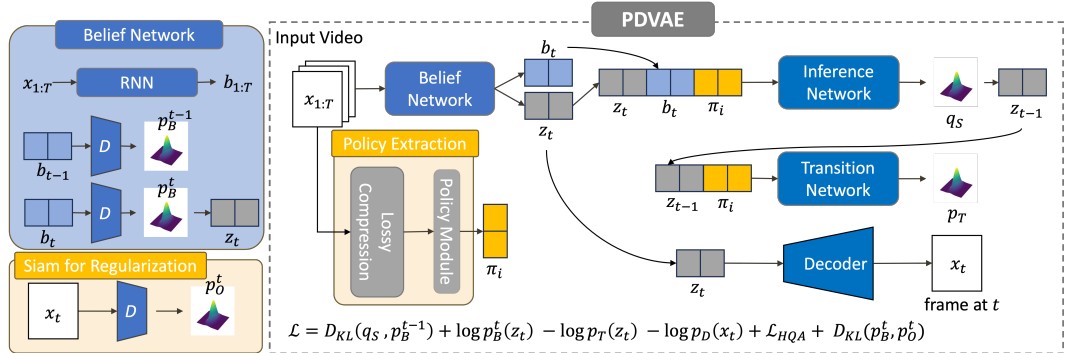

Figure 3: **Training Procedure.** PDVAE extends TD-VAE by incorporating modules highlighted in yellow for policy extraction and regularization using the Siamese architecture, which results in the inclusion of extra terms in the loss function: $\mathcal{L}_{HQA}$ and $D_{KL}(p_B^t \| p_O^t)$). For the succinctness, we have denoted the belief distributions $p_B(z_t|b_t)$ as $p_B^t$, $p_B(z_{t-1}|b_{t-1})$ as $p_B^{t-1}$, and $D$ for the D Block in Appendix . The detailed model architectures are in the appendix.

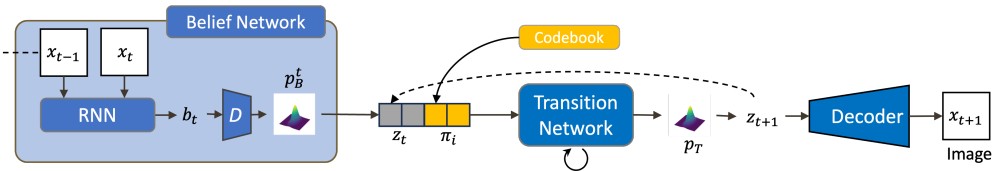

Figure 4: **Generation Procedure**. The state is sampled with the input video and the policy is selected by the user. The transition network takes the state and policy to generate $p_T$. The state $z_{t+1}$ is sampled from $p_T$ and again passed to the transition network with the policy for the roll-out. The sequence of states are decoded to video with decoder.

Secondly, the regularization is introduced to the belief $b_t$ through the incorporation of a Siamese architecture (Chopra et al., 2005; Koch et al., 2015). The model encodes the state $z_t$ with the input video $x_{1:t}$ using the forward RNN as illustrated on the belief network of Figure 3, so the information related to the policy is encapsulated in the state. To distill the policy-related information in the state, we add regularization to the belief with the Siamese architecture. We utilize the single observation $x_t$ to regularize the belief, since the single observation does not contain policy-related information. Using the same $D$ block, which encodes a state $z_t$ to belief $p_B^t$, we encode the observation $x_t$ to $p_O^t$, and add $D_{KL}(p_B \| p_O)$ to the Equation 4 as following:

$$\mathcal{L}^{\text{PDVAE-reg}} = \mathbb{E}_{\substack{z_t \sim p_B(z_t|b_t) \\ z_{t-1} \sim q_S(z_{t-1}|z_t, b_t, b_{t-1}, \pi)}} \Big[ \log p_D(x_t|z_t) - \log p_B(z_t|b_t) + \log p_T(z_t|z_{t-1}, \pi) \quad (6)$$

$$+ \log p_B(z_{t-1}|b_{t-1}) - \log q_S(z_{t-1}|z_t, \pi, b_t, b_{t-1}) \Big]$$

$$+ \mathcal{L}^{\text{HQA}} + D_{\text{KL}}(p_B(z_t|b_t) \| p_O(z_t|x_t))$$

### 3.4 TRAINING AND GENERATION PROCEDURES

We first elaborate on the training procedure of PDVAE as illustrated on the Figure 3. We provide the detailed structure of each network in the appendix. The model encodes the video into a code vector $\pi_i$ with the lossy compression module and the policy module, retrieving the ingredients for $\mathcal{L}^{\text{HQA}}$. Then, the model encodes the video again into a sequence of belief states $b_{1:T}$ with the forward RNN. PDVAE randomly selects $t \in [1, T]$ and obtains the distributions: $p_B^{t-1}, p_B^t, p_O^t, q_S, p_T$. Note that the same D block used to gain $p_B^t$ and $p_B^{t-1}$ is used to obtain the $p_O^t$, the distribution for the regularization purpose, with a frame of the video. The state is sampled from $p_B^t$ and $q_S$ and the rest of the distributions are used for the loss of the model. PDVAE tries to match the $(q_S, p_B^{t-1})$ pair and $(p_T, p_B^t)$ pair with KL divergence and log probability, respectively.

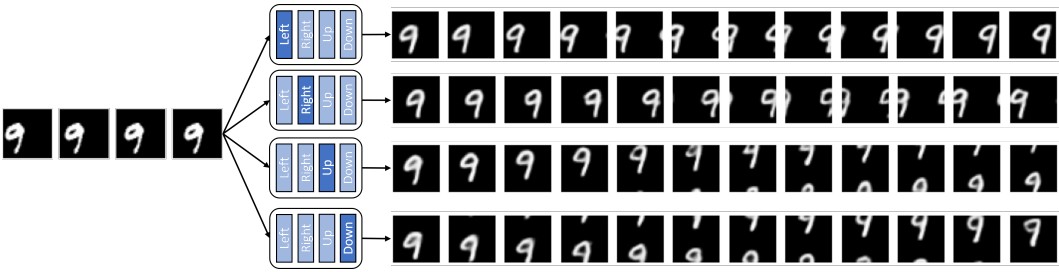

Figure 5: **Different predictions by policy.** PDVAE generates different outcomes (right) given the same input observation (left) depending on the choice of policy. The box represents the codebook of the policy module where the code vector is annotated with the direction.

During the generation, PDVAE employs only the belief network, transition network, decoder, and codebook from the policy module as illustrated in Figure 4. Given an arbitrary length of a video, the belief network encodes the video into a state. The policy is selected by the user and is concatenated with the state for the input of the transition network. The model generates video in an autoregressive manner, by passing the state sampled from the $p_T$ into the transition network again. The decoder decodes the sequence of states to the video.

# 4 RELATED WORK

Recent progress in the deep generative model has led to the advancement on the video generation models. Broadly speaking, the video generation model can be categorized into unconditional video generation or conditional video generation. The former one aims to generate any video that follows the distribution of training dataset (Vondrick et al., 2015; Saito et al., 2020). The conditional video generation can be categorized by the type of conditioning signal.

Among the conditional video generation models, the video prediction problem has been widely studied (Mathieu et al., 2015; Finn et al., 2016; Babaeizadeh et al., 2017; Gregor et al., 2019; Kwon & Park, 2019; Franceschi et al., 2020). The objective of video prediction is to generate future frames given the past frames where the generated frames are spatiotemporally consistent. Earlier works (Mathieu et al., 2015; Finn et al., 2016) have modeled the prediction with the deterministic predictive models, which are unable to account for the inherent stochasticity of the real-world video. In order to integrate the stochasticity, several methods employed the GAN (Kwon & Park, 2019) and VAE formulations (Babaeizadeh et al., 2017; Gregor et al., 2019; Franceschi et al., 2020). These methods are able to generate diverse frames which are consistent to input frame spatially and temporally. Inspired from TD-VAE (Gregor et al., 2019), we additionally condition the policy to generate diverse frames aligned to the policy.

The conditional video generation model can control the generation with the action label (Kim et al., 2020; Menapace et al., 2021). *GameGAN* (Kim et al., 2020) is proposed as the neural simulator of a game. GameGAN generates the gameplay video accordingly to the user's keyboard action input. During the training, the model takes sequence of frames and keyboard actions to learn the dynamics of game environment conditioned to the action label. Menapace et al. (2021) has proposed a model *CADDY* for the playable video generation, with which the user can interact with the video generation by giving the action label. CADDY learns a set of distinct action from the real-world video without the label information. The model learns the action space by maximizing the mutual information between embeddings of encoded features of and ground-truth of consecutive frames. Our approach also learns the latent feature that controls the frame-by-frame generation without the label information.

The semantic feature that is most similar to the policy is the categorical dynamics of *MoCoGAN* (Tulyakov et al., 2018). The model assumes the dynamics in the video can be categorized into discrete action such as walking or running and generates video conditioned to the categorical action signal. The model can be trained without the label information by adopting the *InfoGAN* learning (Chen et al., 2016). Similar to our model, MoCoGAN assumes the number of discrete action $K$ is known and learns the categorical action label with the one-hot vector.

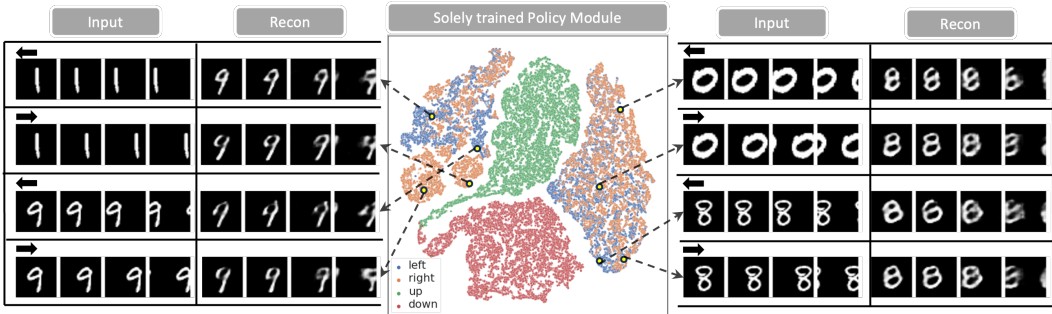

Figure 6: **Policy space learned independently of PDVAE**. Each point represents a video where the color indicates the direction, which is annotated on top of the video in the input section. The video is reconstructed using a single code vector from the policy module trained independently of PDVAE. The dotted arrow points out the location of the video in the embedding space. The policy module categorizes the video with left or right directions into the shape of the digit rather than direction.

## 5 EXPERIMENT

In this section, we provide the evaluations of our method on three datasets: Moving MNIST, KTH action (Schuldt et al., 2004), and VizDoom (Kempka et al., 2016). With the Moving MNIST, we demonstrate that PDVAE learns to distinguish the policy and generates a video aligned with the specified policy along with the ablation of the role of TD-VAE as the regularizer for the policy module. With KTH, we show that the model can alter the policy during the generation which exhibits the smooth transition in change of the policy. With VizDoom, we demonstrate the PDVAE's capability of a neural simulator in the reinforcement learning domain, by generating diverse videos aligned with the specified policy. We provide the experimental setup and qualitative analysis on the generated frames conditioned to the policy for each dataset, followed by the quantitative evaluation compared to the baseline. The choice of hyper-parameters and the detailed model architectures are in the appendix.

### 5.1 MOVING MNIST

We consider a video of 20 frames where a digit from MNIST moves two pixels at each time step in the chosen direction (left, right, up or down). We consider the digit as the agent and the direction as the policy. In this setup, the policy constantly outputs the direction of the movement. The height and width of the video are 32, so the pre-trained module for the lossy compression is stacked four times. We set the codebook slot of the policy module to be four, equal to the number of directions. We consider the digit to be an agent and the direction to be the policy.

Figure 5 demonstrates that PDVAE generates the video in accordance with the specified policy, where the digit's shape remains consistent while the movement direction aligns with the policy. We have annotated the code vector in Figure 5 with the label that appears most frequently in the quantized video. The code vector from the policy modules contains the information related to the policy, whereas the state encapsulates relevant information for prediction except the policy. PDVAE generates videos in which only the direction changes when the conditioned policy differs from the policy of the input video: the lower three rows of the Figure 5. The policy is explicitly disentangled from the state, as the alternation of the policy during the generation only affects the movement of the digit.

We demonstrate the role of the TD-VAE as the regularizer for the policy module with Figure 6. We provide two inductive biases to the model for the extraction of the policy: (1) the constraint to the codebook slots of the policy module and (2) the usage of the policy in the state transitions. Figure 6 depicts the policy space of the policy module without the second inductive bias. With the first inductive bias, the module categorizes the video into four different categories, but not by the policy. Instead, the policy module distinguishes videos of left and right direction by the shape of the digit, which reflects that the policy module preserves the digit-related information to the code vector. Both inductive biases are necessary to extract the policy from the video. We present an extended version of Figure 6 in the appendix, featuring additional digits and directional information. Furthermore,

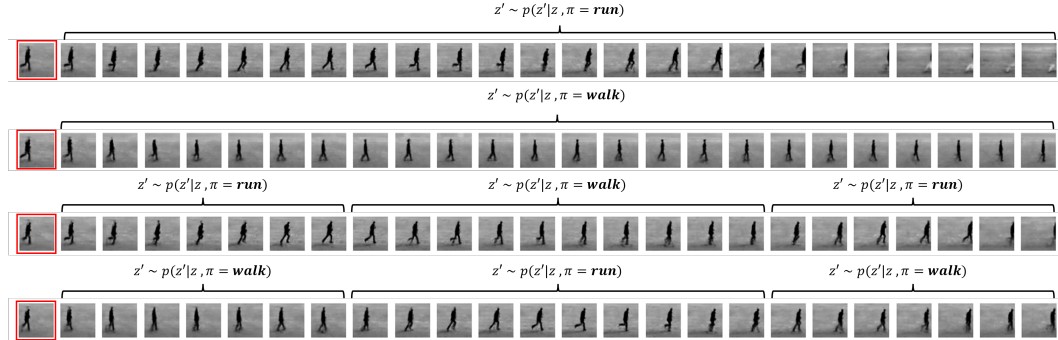

Figure 7: **Smooth transition on the change of the policy**. Four different states from the same $p_B$, the decoded version is marked with red perimiter, and are rolled out with policy indicated above the sequence. Each agent demonstrates the smooth transition between the change in the policy.

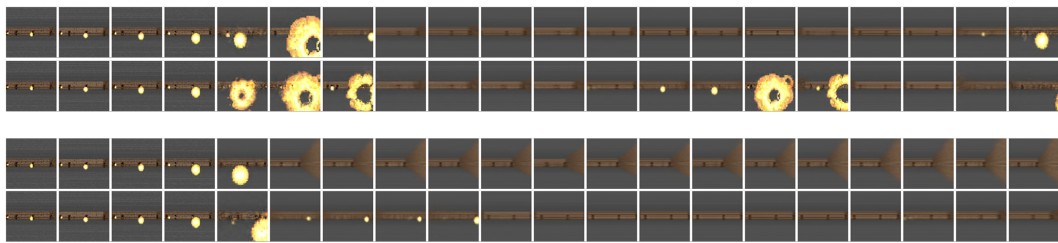

Figure 8: **Diverse simulations by the policy**. The first four frames of the videos in each row are the input to PDVAE. The video on the top two rows are generated with the policy *facing fire* whereas the bottom two are generated with the policy *avoiding*.

we include video reconstructions using the policy module trained alongside PDVAE, along with a scatterplot, all of which can be found in the appendix.

## 5.2 KTH ACTION

The KTH dataset is comprised of videos featuring an individual carrying out a single action in four different backgrounds. We have selected a video of a person either running or walking in the outdoor background, where the person's action remains unchanged throughout each video. We consider the person in the video as an agent and the action as the policy. For the pre-processing, we center crop and set the resolution to 64x64. The lossy compression module is stacked five times and the The lossy compression module is stacked five times and the codebook slots for the policy module are set to two. We have added the cross entropy loss with the pseudo label to the loss from the pre-trained policy module to enhance the generation quality of generation (Lee et al., 2013).

The Figure 7 depicts generations of four distinct videos, all based on the same input frames but conditioned on different sets of policies. These policies are annotated based on their most frequent occurrence in the quantized video data. The upper two rows of the Figure 7 are generated with the constant policy. Although the input sequences are the same, the agent's actions differ according to the conditioned policy. In contrast, the lower two rows of generated video employ alternating policies. When the policy shifts (e.g., from "run" to "walk"), the agent within the generated video gradually adjusts its behavior to align with the new policy. The state encapsulates the information on the environment and the current status of an agent. Moreover, the transition process, guided by the policy, effectively reflects the agent's status, resulting in a seamless transition in video when policies change. By alternating policies, PDVAE is capable of generating videos that are not that are not present in the training dataset.

## 5.3 VIZDOOM

The video from the VIZDOOM-TAKECOVER environment contains the egocentric view of a person in an enclosed room, with the person serving as the agent. In this environment, devils randomly appear and shoot fireballs at the agent, who can take an action from (move left, move right, or stay) on each time step. We have created 10k episodes in which random actions were taken at each time step. These episodes are categorized as "avoiding" if the agent's health remains intact and "facing fire" if

Table 1: Comparison with the baseline on all dataset. $\pi$-Acc in %. The MoCoGAN on the first row is trained with the label information and the MoCoGAN- on the second row is trained without the label.

| Model | MNIST | | | KTH | | | vizdoom | | |
|---|---|---|---|---|---|---|---|---|---|
| | FID $\downarrow$ | FVD $\downarrow$ | $\pi$-Acc $\uparrow$ | FID $\downarrow$ | FVD $\downarrow$ | $\pi$-Acc $\uparrow$ | FID $\downarrow$ | FVD $\downarrow$ | $\pi$-Acc $\uparrow$ |
| MoCoGAN | 24.6 | 282 | 98.8 | 61.9 | 511 | 70.4 | 14.3 | 903 | 69.8 |
| MoCoGAN- | 24.2 | 391 | 24.8 | 70.5 | 604 | 50.4 | 11.8 | 587 | 50.7 |
| PDVAE | 8.63 | 64 | 99.6 | 50.2 | 449 | 72.7 | 27.0 | 637 | 73.8 |

the agent's health is diminished. These categories, "avoiding" and "facing fire", serve as the agent's policies. We have stacked five times for the lossy compression module and the input video for the module is converted to greyscale videos.

PDVAE produces videos featuring diverse action trajectories, with the agent adhering to the specified policy, as depicted in the Figure 8. For instance, when the "avoiding" policy is conditioned to the transition, the agent takes different actions, with one moving right and another left, yet both successfully avoid the fireballs. In the generated video of top two rows, the agents initially perform similar actions, resulting in them getting hit by the fireballs, but their subsequent actions differ. Given the state and the policy, the model obtains the distribution over a future state with the transition network and samples the state to move forward, where the state transition is characterized by the policy. The innate stochasticity of the state transition enables PDVAE to generate diverse videos but an agent within the video acts according to the conditioned policy.

### 5.4 QUANTITATIVE EVALUATION

We evaluate the generated video with quantitative metrics and compare it to the baseline we select.

**Video Quality** We evaluate the quality of the generated video with FID (Heusel et al., 2017) and FVD (Unterthiner et al., 2018). We have generated 16 frames of video conditioned to the designated policy and calculated the score with the ground truth video of the respective policy. The test set is considered for the calculation of both metrics. We report the average FVD score from each policy. We calculate the average FID of 16 frames from each policy and report the average of it.

**Policy Metric** We introduce $\pi$-Acc to evaluate the quality of policy space. This metric measures how well the generated frames follow the conditioned policy. To this aim, we train a linear or convolutional classifier with the label. We have generated 16 frames of video to obtain the metric. We report the $\pi$-Acc measured on the generated video from the test set.

**Baseline Selection** Since we are the first to extract the policy from the video and generate frames conditioned to the policy, we have selected a model that learns the closest resemblance of the policy. As mentioned in the section 4, the MoCoGAN shares a similar latent feature to ours, known as Categorical Dynamics, which influences the motion trajectories of the entire video, mirroring our policy. We have conducted evaluations of the MoCoGAN with and without label information, yielding quantitative metrics for video quality and the policy space, as displayed in Table 1. The first row presents results for MoCoGAN trained with label information, while the second row depicts outcomes for MoCoGAN trained without label information.

## 6 CONCLUSION

In this paper, we present PDVAE to generate videos aligned with the specified policy by a user, by learning to extract the policy from the video without label information and to model the transition conditioned to the policy. PDVAE can generate diverse scenarios aligned with the conditioned policy where the user can alter the policy during the generation. We have derived a novel ELBO to learn such disentangled representation along with the architecture and training procedure for the model. The experiments with three datasets, Moving MNIST, KTH action dataset, and VizDoom, demonstrate the capability of PDVAE. PDVAE uses basic neural network architecture such as convolutional LSTM, convolutional layer, and multi-layer perceptron with one or two residual layers. Compared to the state-of-the-art video generative models, PDVAE consists of simple architecture, which restricts us from performing experiments in rather smaller size of videos. In future works, to overcome the limitation of PDVAE, we aim to find suitable neural architecture, rather than basic ones, to test our model with the more complex dataset. We also explore several potential applications in reinforcement learning, with the simulation results generated with PDVAE.

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
