})}} \left[ \log \frac{p(z_{t-1},z_t,\pi,x_t|x_{<t})}{q(z_{t-1},z_t,\pi|x_t,x_{<t})} \right]$$

$$\geq \mathop{\mathbb{E}}_{\substack{(z_{t-1},z_t,\pi)\sim \\ q(z_{t-1},z_t,\pi|x_{\leq t})}} \left[ \log p(z_{t-1},z_t,\pi,x_t|x_{<t}) - \log q(z_{t-1},z_t,\pi|x_t,x_{<t}) \right]$$

$$= \mathop{\mathbb{E}}_{\substack{(z_{t-1},z_t,\pi)\sim \\ q(z_{t-1},z_t,\pi|x_{\leq t})}} \left[ \log p(x_t|z_{t-1},z_t,\pi,x_{<t}) + \log p(z_{t-1},z_t,\pi|x_{<t}) - \log q(z_{t-1},z_t,\pi|x_{\leq t}) \right]$$

(since $x_t$ only depends on $z_t$)

$$= \mathop{\mathbb{E}}_{\substack{(z_{t-1},z_t,\pi)\sim \\ q(z_{t-1},z_t,\pi|x_{\leq t})}} \left[ \log p(x_t|z_t) + \log p(z_{t-1},z_t,\pi|x_{<t}) - \log q(z_{t-1},z_t,\pi|x_{\leq t}) \right]$$

$$= \mathop{\mathbb{E}}_{\substack{(z_{t-1},z_t,\pi)\sim \\ q(z_{t-1},z_t,\pi|x_{\leq t})}} \left[ \log p(x_t|z_t) + \log p(\pi|x_{<t}) + \log p(z_{t-1}|\pi,x_{<t}) + \log p(z_t|z_{t-1},\pi,x_{<t}) \right.$$
$$\left. - \log q(z_{t-1},z_t,\pi|x_{\leq t}) \right]$$

(since $z_t$ only depends on $z_{t-1}$ and $\pi$)

$$= \mathop{\mathbb{E}}_{\substack{(z_{t-1},z_t,\pi)\sim \\ q(z_{t-1},z_t,\pi|x_{\leq t})}} \left[ \log p(x_t|z_t) + \log p(\pi|x_{<t}) + \log p(z_{t-1}|\pi,x_{<t}) + \log p(z_t|z_{t-1},\pi) \right.$$
$$\left. - \log q(z_{t-1},z_t,\pi|x_{\leq t}) \right]$$

$$= \mathop{\mathbb{E}}_{\substack{(z_{t-1},z_t,\pi)\sim \\ q(z_{t-1},z_t,\pi|x_{\leq t})}} \left[ \log p(x_t|z_t) + \log p(\pi|x_{<t}) + \log p(z_{t-1}|\pi,x_{<t}) + \log p(z_t|z_{t-1},\pi) \right.$$
$$\left. - \log q(\pi|x_{\leq t}) - q(z_t|\pi,x_{\leq t}) - q(z_{t-1}|z_t,\pi,x_{\leq t}) \right]$$

(since $z_{t-1}$ (and $z_t$) only depends on past history $x_{<t}$ (and $x_{\leq t}$))

$$= \mathop{\mathbb{E}}_{\substack{(z_{t-1},z_t,\pi)\sim \\ q(z_{t-1},z_t,\pi|x_{\leq t})}} \left[ \log p(x_t|z_t) + \log p(\pi|x_{<t}) + \log p(z_{t-1}|x_{<t}) + \log p(z_t|z_{t-1},\pi) \right.$$
$$\left. - \log q(\pi|x_{\leq t}) - q(z_t|x_{\leq t}) - q(z_{t-1}|z_t,\pi,x_{\leq t}) \right]$$

The last line of the equation is the Equation 3 which is the ELBO of the PDVAE. We have changed the sequence of the terms for better understanding of ELBO in Equation 3.

# B  MODEL ARCHITECTURE

In this section, we illustrate the detailed architecture of PDVAE where the choice of parameters can be found on Appendix C. The notations used to describe the modules are as follows:

**Linear(a)**: Linear layer with the output dimension **a**

**Conv2D(a,b,c,d)**: 2D-Convolution layer with output channel size **a**, kernel size **b**x**b**, stride size **c**, and padding size **d**.

**ConvLSTM(a,b,c,d)**: 2D Covolutional LSTM layer with output channel size **a**, kernel size **b** x **b**, stride size **c**, and padding size **d** Shi et al. (2015)

**Mish**: Mish activation function Misra (2019)

**Upsample(a)**: Upsample the height and width of an input image or image sequence with the nearest neighbor interpolation of scaling factor **a**

**Reshape(a)** Reshape the input to output size of **a**, which can be a tuple or an integer

## B.1 POLICY MODULE

Each layer of the lossy compression module and the policy module consists of an encoder, a vector quantization layer, and a decoder, where the architecture for encoder and decoder are illustrated on Table 2. The encoder and decoder are stacked up with the following layers as a feed-forward network where $d_E$ and $d_D$ represent the dimensions of the encoder and decoder. The decoder of the bottom layer of the lossy compression module has an additional sigmoid function for the reconstruction of the image. The inputs for encoders except one from the bottom layer are normalized with running statistics for the stable training as HQA does. The code vector of the VQ layer codebook has the dimension of $d_\pi$.

In order to compress the video to a 1D vector, we take the sequences of images as the input for the lossy compression module. $T$ indicates the length of the sequences. We have used the grayscale images instead of RGB images as the policy is irrelevant to the channel, hence the input shape for the lossy compression module $H \times W \times T$ where $H$ and $W$ are the height and width of the video. The policy module takes the encoded video from the lossy compression module with shape $2 \times 2 \times d_\pi$

| Encoder | Decoder |
|---------|---------|
| Conv2D($d_E$/2, 3, 2, 1) | Conv2D($d_D$, 3, 0, 1) |
| Mish | Upsample(2) |
| Conv2D($d_E$, 3, 0, 1) | Conv2D($d_D$/2, 3, 0, 1) |
| Mish | Mish |
| Conv2D($d_\pi$,1,0,0) | Conv2D($ts$, 3, 0, 1) |

Table 2: Encoder and Decoder architecture of the Policy Extraction Module

## B.2 MODULES FOR PDVAE

We elaborate on the networks used in PDVAE except for the lossy compression module and the policy module. As TD-VAE utilized the hierarchical structure, we have used the same hierarchical structure of the state space model. We have used the same $D$ block from TD-VAE, which outputs a normal distribution, and modified the block as $HB$ to output a vector instead, illustrated in Figure 9. We have used Resblock as depicted in Figure 9 in the decoder of PDVAE.

The Table 3 illustrates the modules for the VizDoom experiment, where the difference with other experiments is the number of ResBlock in the decoder and the existence of the $HB$ Block. Both experiments does not have the $HB$ block to resize the state, the Moving MNIST experiment does not use the ResBlock, and the KTH action experiment uses a single ResBlock.

In the Table 3, D and HB indicate the $D$ Block and $HB$ block from the Figure 9. $d_b$, $d_z$, and $d_\pi$ stands for the dimension of the belief, state, and policy, respectively. $d_x$ is the multiplication of the height and width of the input image sequence, and $C_x$ stands for the channel of the input image. $C_h$ is the hidden size of the channel in the ConvLSTM. $N_h$ indicates the number of hierarchy (stack) of the PDVAE and $h$ represents the size of hidden units. We have employed the same hierarchical (stacked) structure of TD-VAE with the addition of the policy $\pi$ in the transition and inference networks.

The transition and inference networks except the ones from the top of the stack additionally takes the input from the upper hierarchy, hence the input size of $D$ block becomes $N_h * d_z + d_\pi + d_z$ and $d_b + N_h * d_z + d_\pi + d_z$, respectively.

The belief network creates the belief state with two ConvLSTM and $HB$ Block. Then, the network samples the state from belief on each hierarchy using the belief state with size $d_b$. The networks except the one from the top level, similarly to transition and inference network, takes the additional input of state sampled from the layer below. The belief network except the top takes $d_b + d_z$ as the input. The Siamese architecture shares $HB$ Block and $D$ block to produce the distribution for the regularization and takes $x_t$ as the input instead of the hidden state from the ConvLSTM.

| Belief Network | Decoder Network |
|---|---|
| ConvLSTM($C_h$, 3, 1, 1) | HB($N_h * d_z$, h, $d_x$/4) |
| ConvLSTM($C_x$, 3, 1, 1) | Reshape(($\sqrt{d_x}$/2,$\sqrt{d_x}$/2)) |
| Reshape(($C_x * d_x$)) | Conv2D($d_b$,3,0,1) |
| HB(($C_x * d_x$), $2 * h$, $d_b$) | ResBlock($d_b$, $d_b$/2) |
| | ResBlock($d_b$, $d_b$/2) |
| D($d_b$, h, $d_z$) | Upsample(2) |
| **Transition Network** | Conv2D($d_b$/2, 3, 0, 1) |
| D($N_h * d_z + d_\pi$, h, $d_z$) | Mish |
| **Inference Network** | Conv2D($C_x$, 3,0,1) |
| D($d_b + N_h * d_z + d_\pi$, h, $d_z$) | Sigmoid |

Table 3: Architecture of the networks used in PDVAE except the Policy Extraction Module

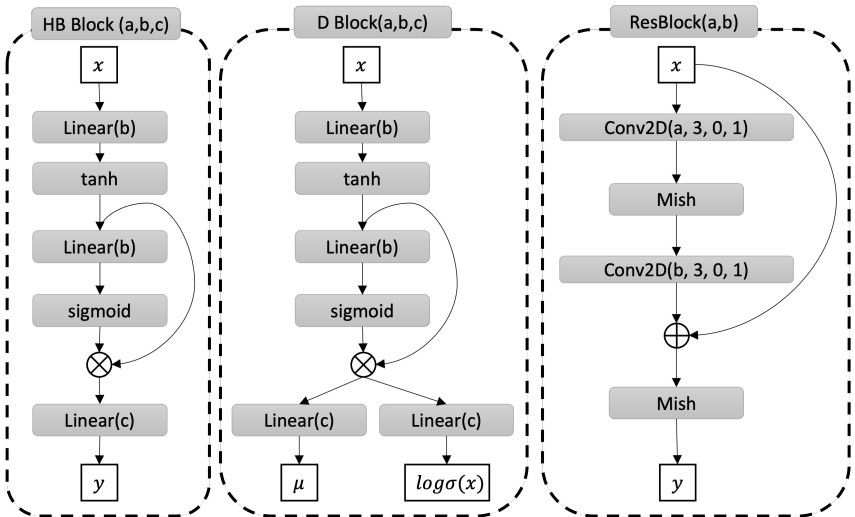

Figure 9: **HB Block(a,b,c)**: given the input $x$ with the size **a**, HB Block outputs a vector $y$, with linear layers with hidden unit size **b** and output size **c**. **D Block(a,b,c)**: given the input $x$ with the size **a**, D Block outputs a normal distribution $[\mu, \log \sigma]$ with linear layers with hidden unit size **b** and output size **c**. **ResBlock(a,b)**: residual, which outputs the same channel to input channel, of convolution layers with hidden channel size **a** and input channel size **b**.

## C PARAMETER CHOICE

We have trained our model with a single GPU, NVIDIA GeForce 3090 RTX.

### C.1 LOSSY COMPRESSION MODULE AND POLICY MODULE

In this section, we illustrate the parameter choice for the lossy compression module and the policy module along with the shape of input video and the preprocess method by dataset. The Table 4 illustrates the parameter choice for each experiment. The codebook slots indicate the number of code vectors in the codebook of VQ layer of the policy extraction module. $d_E$ and $d_D$ consist of the hidden channel size of the encoder and decoder of each layer where the leftmost is the parameter for the bottom layer and the rightmost is the parameter for the top layer. We have changed the image sequence to grayscale for the KTH action and VizDoom Experiment. We have used the timesteps of the image sequence as the channel of an image for all experiments. Therefore, the tuple in the input size stands for the height, width, and length of the sequences of images.

**Preprocess** For the KTH action experiment, we have adjusted the contrast by a factor of two to the images to highlight the action of a person more. For the input, we have used the difference

between frames as the behavior of a person becomes more explicit, hence the length of the sequence becomes 19. For the VizDoom experiment, we sample the episodes with ScreenFormat of RGB24, ScreenResolution of RES_200X125, and skip rate of 3. We have resized the image of 200x125 to height 64 and cropped the center of the image to make a 64x64 square image. We have normalized the video with pixel values between [0,255] to [0,1].

|  | **Moving MNIST** | **KTH action** | **VizDoom** |
|---|---|---|---|
| Input shape | (32,32,20) | (64,64,19) | (64,64,20) |
| Batch size | 2048 | 128 | 128 |
| $d_E$ | [96, 96, 192, 288, 480] | [96, 96, 192, 288, 480, 960] | [96,96, 192, 288, 480, 960] |
| $d_D$ | [128, 128, 256, 384, 640] | [96, 192, 288, 480, 768, 1536] | [160, 320, 480, 800, 1280, 2560] |
| $d_\pi$ | 128 | 256 | 512 |
| Codebook Slots | [256,256,256,256,4] | [256,256,256,256,2] | [256,256,256,256,2] |
| Epoch | [150,150,150,150,150] | [200,200,200,200,200] | [200,200,200,200,200] |
| learning rate | 4e-4 | 4e-4 | 4e-4 |

Table 4: Parameter choice of the Policy Extraction Module

## C.2 PDVAE

In this section, we illustrate the parameter choice for the networks in PDVAE with Table 5. PDVAE takes video as an input, hence the input shape (ts, H, W, C) represents the length, height, width, and channel of the sequences of images. For KTH, we have added the cross entropy loss with the pseudo label from the pre-trained policy module to enhance the generation quality (Lee et al., 2013).

**Preprocess** We have not done any preprocess to Moving MNIST. For KTH action, we adjust the contrast by a factor of two for the input and use the original frames, not the difference. We have applied the same prerprocess as the one from the training of the lossy compression module and policy module to the Vizdoom except for the conversion of grayscale. We have used the RGB video for the PDVAE.

|  | **Moving MNIST** | **KTH action** | **VizDoom** |
|---|---|---|---|
| Input shape | (20,32,32,1) | (20, 64,64, 1) | (20,64,64,3) |
| Batch size | 128 | 128 | 32 |
| $d_x$ | 1024 | 4096 | 12288 |
| $C_x$ | 1 | 1 | 3 |
| $C_h$ | 1 | 3 | 50 |
| $d_b$ | 256 | 1024 | 512 |
| $d_z$ | 128 | 512 | 64 |
| $d_\pi$ | 128 | 256 | 512 |
| $h$ | 1024 | 2048 | 2048 |
| $N_h$ | 2 | 2 | 4 |
| $\beta$ | 1 | 1 | 1 |
| Epoch | 6000 | 10000 | 10000 |
| learning rate | 4e-4 | 2e-4 | 1e-4 |

Table 5: Parameter choice of the networks in PDVAE

## D  ADDITIONAL EXPERIMENT RESULTS ON MOVING MNIST

In this section, we present additional experiment results on the Moving MNIST data.

As one can observe, the reconstructed images from the top layer are similar across the digit within the same direction. The policy extraction module has compressed the sequences of images and learns the representation of the policy.

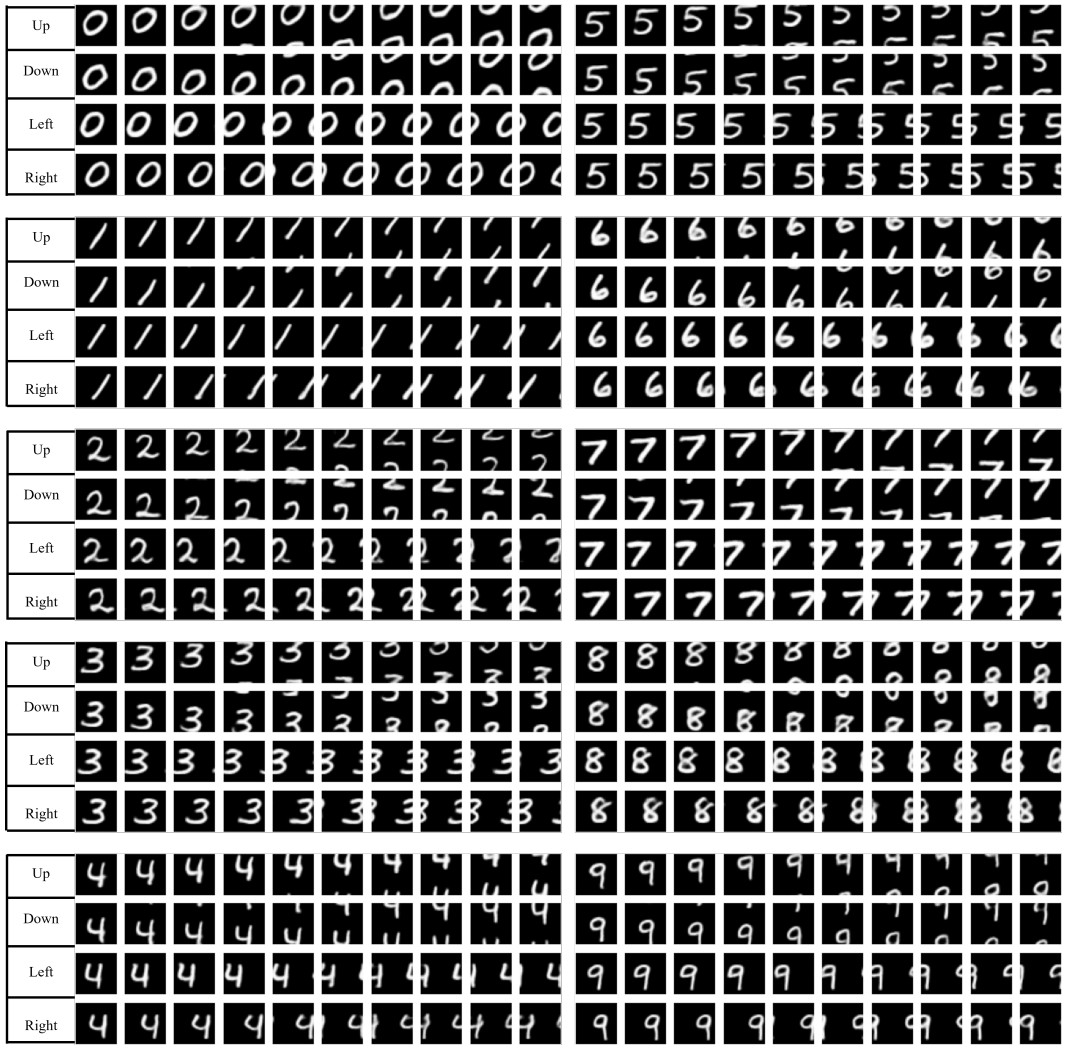

Figure 10: Different choice of policy for the generation of every digit

Figure 13 demonstrates the generated image sequence of every digit for every direction. This figure is the extended experiment results of the Figure 5. Given any input sequence, PDVAE can alter the policy of an agent and generates accordingly to the policy without changing the shape of the digit.

## E    ADDITIONAL EXPERIMENT RESULTS ON VIZDOOM

In this section, we present additional experiment results on the Vizdoom data. Figure 13 demonstrates the generated image sequence of every digit for every direction. This figure is the extended experiment results of the Figure 5. Given any input sequence, PDVAE can alter the policy of an agent and generates accordingly to the policy without changing the shape of the digit.

## F    COMPARISON WITH TD-VAE

In this section, we provide the quantitative comparison between TD-VAE (Gregor et al., 2019) and PDVAE. PDVAE is the TD-VAE with the lossy compression module, policy module, and the siam architecture. TD-VAE for the comparison is trained with the same hyperparameters of PDVAE. Since TD-VAE encodes the complete information for prediction, including policy, to the state $z_t$, TD-VAE performs better on the generation quality. However, the difference in FVD and FID are not not

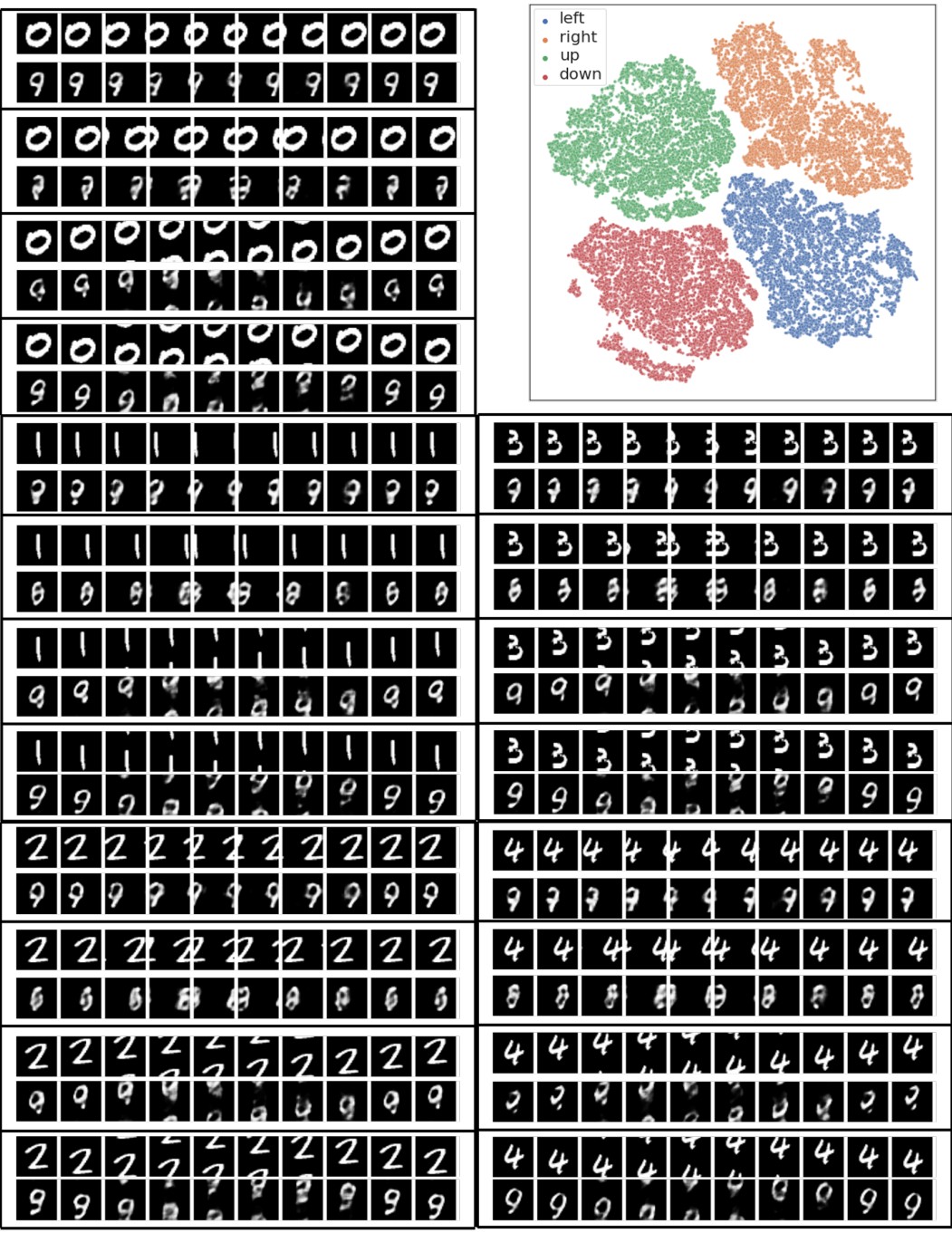

Figure 11: The scatter plot indicates the policy module trained with PDVAE distinguishes the videos into four categories by the policy. Input video and the reconstruction of the video from the policy module in PDVAE using a single code vector. Each row in the table represents the input video and the reconstruction video from the policy module where the upper one is the input and lower one is the reconstruction. The reconstructions of different digits with the same digit are analogous. The policy module trained with PDVAE preserves only the policy-related semantic feature in the code vector.

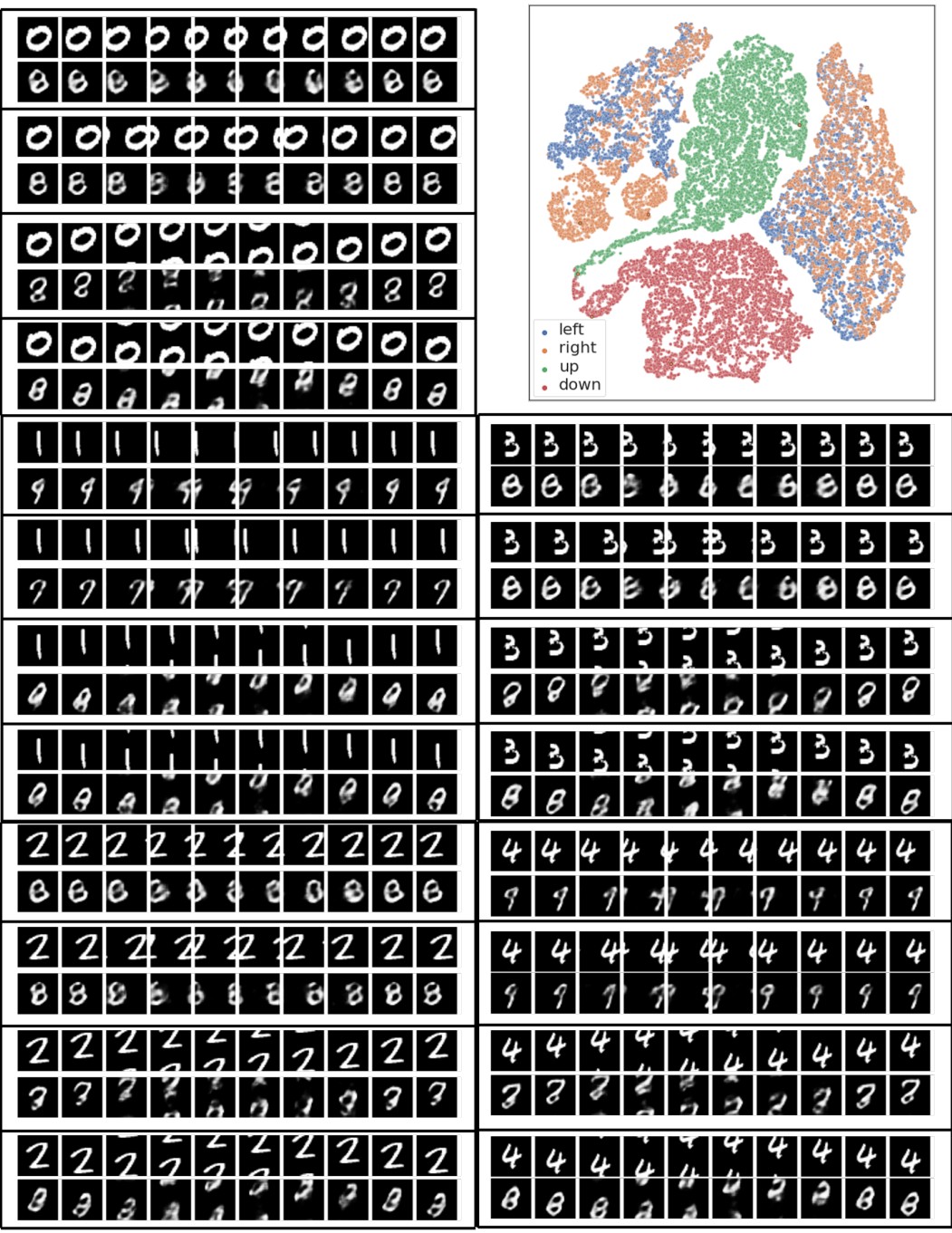

Figure 12: The scatter plot indicates the policy module trained independently of PDVAE failed to distinguish the videos by the policy. Input video and the reconstruction of the video from the policy module trained independently of PDVAE using a single code vector. Each row in the table represents the input video and the reconstruction video from the policy module where the upper one is the input and lower one is the reconstruction. The reconstructions of different digits with the same digit are not analogous.

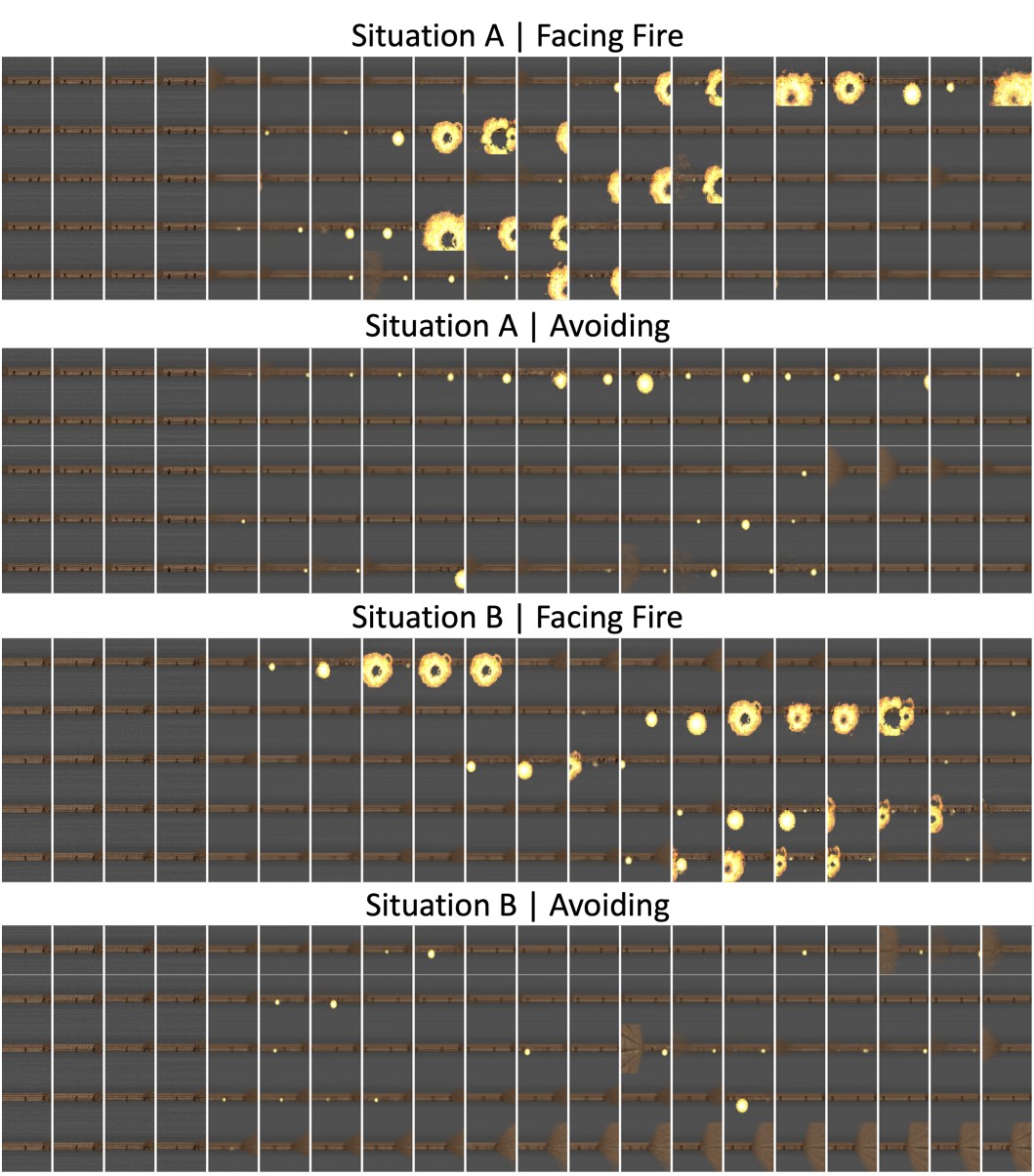

Figure 13: Each row represents a generated sequence with the policy and situations (the first four images). Given the same situations, the agent acts according to the policy. However, even with the same policy and situation, the generated videos shows diverse outcomes because of the stochasticity within the model. The above figures demonstrate the capability of PDVAE over the question on "what if someone with different intention/policy/strategy will do in a given situation?"

Table 6: Quantitative comparison between PDVAE and TD-VAE

| Model | MNIST | | KTH | | vizdoom | |
|---|---|---|---|---|---|---|
| | FID ↓ | FVD ↓ | FID ↓ | FVD ↓ | FID ↓ | FVD ↓ |
| TD-VAE | 7.99 | 53 | 57.8 | 364 | 26.5 | 621 |
| PDVAE | 8.63 | 64 | 50.2 | 449 | 27.0 | 637 |

meaningful enough in Moving MNIST and VizDoom. As Figure 7 indicates, it takes a few frames for the person to switch the motion from one another. The generated video does not exist in the original dataset, hence the FVD score of PDVAE is lower than the score of TD-VAE. Despite the lower performance, PDVAE can control the generation according to the policy, which can be potentially utilized to gaming, video, and robotics tasks.