# OpenReview forum: "Policy Disentangled Variational Autoencoder"
_ICLR.cc/2024/Conference — Submitted to ICLR 2024_

### Official Review · Reviewer_FTr9 · 2023-10-27

**Soundness:** 2 fair
**Presentation:** 3 good
**Contribution:** 1 poor
**Rating:** 3
**Confidence:** 4

**Summary:**

This paper introduces a discrete latent code into the TD-VAE framework to better model discrete actions in videos and disentangle the actions from states. Experiential results on MNIST, KTH, and Vizdoom show that their model, policy-disentangled VAE, can recognize the discrete action in the input videos and generate the futures conditioned on other actions.

**Strengths:**

- The paper is easy to understand.

**Weaknesses:**

- The method evaluation is not convincing. As a 2024 submission on video generation, it only tests on three easy benchmarks: Moving MNIST, KTH with only one type of video (walk/run), and Vizdoom. In addition, the baseline method it compared with is MoCoGAN in 2018. This evaluation setting is definitely not at the bar of 2023. Even papers from three years ago like Dreamer [1] (ICLR 2020) have a more difficult setting.
- The contribution is limited. Although the method is named policy-disentangled VAE, it is not conditioned on “policy”, but discrete actions like move left or move right. This can be achieved in papers 5 years ago like World Models [2] (NIPS 2018).

- In addition, the idea of introducing discrete modes in VAE for better future modeling is not new and has been explored in other fields at least 3 years ago. (e.g., Trajectron++ [3] (ECCV 2020) in motion forecasting)


[1] Hafner, Danijar, et al. "Dream to control: Learning behaviors by latent imagination." arXiv preprint arXiv:1912.01603 (2019).

[2] Ha, David, and Jürgen Schmidhuber. "World models." arXiv preprint arXiv:1803.10122 (2018).

[3] Salzmann, Tim, et al. "Trajectron++: Dynamically-feasible trajectory forecasting with heterogeneous data." ECCV 2020

**Questions:**

In general, I think the evaluation setting is too easy for 2023. Showing performance that is compatible with Dreamer v3 [4] (2023) or even Dreamer (2020) on tasks they evaluate will be a plus.

[4] Hafner, Danijar, et al. "Mastering diverse domains through world models." arXiv preprint arXiv:2301.04104 (2023).

---

> ### Author Response · Authors · 2023-11-21
>
> Thank you for taking the time to read and organize the review of our works. We appreciate your concerns about our proposed method and experiments. Following are the statements on the weakness of our paper. Please refer to the response to all reviewers for our response on contribution, baseline, and evaluation tasks.
>
> The objective of our paper is not to generate high-quality video. Our argument and contribution are not to generate high-quality video while controlling the policy of an agent in the video. The contribution of the paper is that PDVAE categorizes the unlabeled video by the policy and generates video according to the policy. The experimental results with three datasets demonstrate the capability of PDVAE to learn the policy representation and controllability.
>
> $\textbf{World model}$ utilizes the VAE architecture to generate images and video. However, the model does not have model architecture nor loss to learn and generate policy. The model learns the impact of action on the dynamics of the environment, overlooking the intention behind the action. We have raised the problem of overlooking in the introduction. $\textbf{Dreamer’s}$ objective is to learn the agent’s model dynamics. The model intends to explicitly learn the model dynamics. According to the objective of PDVAE, the model does not need to learn the dynamics explicitly. PDVAE learns the dynamics implicitly with the generative model. $\textbf{Trajectron++}$ learns the discrete latent feature $z_{mode}$ for better future modeling where PDVAE learns the discrete latent feature $\pi$ from the video and uses it to simulate diverse scenarios conditioned to the $\pi$. Trajectron++ does not control the generation with the latent mode. Considering the objective and methods used, we believe each model and PDVAE in different research criteria.
>
> Thank you.

---

> > ### Comment · Reviewer_FTr9 · 2023-11-22
> >
> > Thank you for the answers. The latent code z in Trajectron++ in 2020 is learned from input sequence and used to simulate diverse future possibilities. PDVAE does the same thing but with simple videos. As we still don't have more recent baselines to compare with and more convincing experiments, and there is no obvious difference between PDVAE and methods like Trajectron++. I decide to keep my score.

---

### Official Review · Reviewer_bJsN · 2023-10-30

**Soundness:** 2 fair
**Presentation:** 2 fair
**Contribution:** 3 good
**Rating:** 3
**Confidence:** 3

**Summary:**

### Problem Statement

The paper addresses the problem of overlooking the underlying intentions or policies driving the actions in videos while utilizing deep generative models for video generation. Traditional models primarily treat videos as visual representations of actions performed by agents without delving into the motivations or intentions behind those actions. This lack of attention towards the intention behind actions can limit the understanding and representation of behaviors exhibited in videos.

### Main Contributions

The main contribution of the paper is the introduction of a novel model called Policy Disentangled Variational Autoencoder (PDVAE). PDVAE aims to learn the representation of the policy, akin to the underlying intention guiding the actions, without supervision, alongside learning the dynamics of the environment conditioned to the policy. Unlike previous models, PDVAE can generate diverse videos aligned with a specified policy, and even allows for altering the policy during the generation to produce varied behavioral outcomes in videos. The model differentiates videos based on the policy of an agent and can generate videos where each agent adheres to the specified policy.

### Methodology and experiments

The PDVAE extends beyond Temporal-difference Variational Autoencoder (TD-VAE) by disentangling the state and policy, thereby providing a more nuanced understanding and representation of agents' behaviors in videos based on their underlying intentions or policies. Through qualitative and quantitative experimental validations on three video datasets (Moving MNIST, KTH action dataset, and VizDoom), the paper demonstrates the effectiveness of PDVAE in capturing and generating videos based on policy representations, opening up new avenues for more intention- and context-aware video generation.

**Strengths:**

This work is essentially an extension to the referred TD-VAE work, introducing so-called "policy" latent variables to condition the generative decoder. The derivation introducing the latent variable by decomposing the ELBO is neat. The resulted architecture is sophisticated with carefully designed loss.

The smooth transition on the change of the policy latent is intriguing in that the generated video respects the dynamics of its content while being controllable.

**Weaknesses:**

### Limited evaluation tasks and baselines

The video generation mothod is evaluated on three datasets and compared to one baseline, which is not persuasive with respect to the versatility and superiority of the method.

### Lack of ablation study

Despite the sophisticated architecture, there is little ablation study analyzing the importance of its various components.

### Writing

Although the article conveys the ideas successfully, I find it sometimes repetitive, not very well organized, and not precise enough. Grammar mistakes also slightly hinder my comprehension.

### Minor
- The "policy" in this work is really a set of learnable embeddings disentangled from the latent space, while the authors' description tends to confuse it with the concept of policy in reinforcement learning, which is a function / distribution over actions. I understand that the learnable embeddings are intended to capture the "policy" which hypothetically controls the "style" or "mode" of the video generation, but it would help readers understand if the authors could further clarify this and differentiate the distinct concepts.
- It would be beneficial if the authors could include the generated videos in supplement materials for better demonstration.
- Please fix the missing space between the caption of Figure 8 and the main text below it.

**Questions:**

- How is the policy space visualized as a 2D scatter in Figure 6?

---

> ### Author Response · Authors · 2023-11-21
>
> We thank you for your time and effort in your review. We appreciate your recognition of the model architecture and the mathematical support. Below is the response to the concerned weakness of the paper and the question. Please refer to the response to all reviewers for the response over the Limited evaluation tasks and baseline.
>
> 1. Lack of ablation study
>
> The model architecture of PDVAE might seem sophisticated, but essentially the model is composed of the policy-conditioned generative model (TD-VAE with policy), the policy extraction module, and the Siamese architecture for the regularization. We have done the ablation study on the impact of the policy-conditioned generative model on the policy extraction module with Moving MNIST and posted the qualitative results on the paper (Figures 6, 11, and 12). As we have illustrated in the second paragraph of Section 3.3, the Siamese architecture is essential to regulate the state. Without the structure, the conditioned policy does not affect the generation as the state contains complete information on the policy and environment.
>
> 2. Minor: Policy
>
> PDVAE uses the latent variable $\pi$ to encapsulate information on the agent’s intention or behavioral goal. Given samples of videos where each video contains an agent with its own strategy or behavioral goal, the videos can be categorized into discrete numbers by the strategies. From the perspective of the video generative model, such latent embedding can be interpreted as the “style” or “mode”. PDVAE implicitly learns the dynamics of the environment with TD-VAE-based architecture, so interpreting the latent variable as the policy is more appropriate. In reinforcement learning, the policy is a function to represent the agent’s behavioral intention. In PDVAE, we use the coded representation for the policy.
>
> Thank you.

---

> > ### Comment · Reviewer_bJsN · 2023-11-23
> >
> > I thank the authors for the reply. After considering the reviews of other reviewers and the responses, I'm not convinced that the work makes a contribution of enough significance. Introducing structures and therefore interpretability and disentanglement to the latent space through a hierarchical probabilistic model is a long existing approach [1] [2], and with the limited evaluation and experiments, it's hard to argue the superiority of the proposed method. I am also skeptical about the validity of the concept of "policy" that the authors have been emphasizing: Both the agents' policy / behavior and the world dynamics contribute to the formation of observed trajectories, and it seems to me rather narrow to attribute the variations of the trajectories to "policy". For example, for observations of a person playing with a bouncing ball, both the person's policy or behavior and the ball's mass and elasticity collectively decide the ball's movement, and I do not think calling the ball's properties "policy" is proper.
> >
> > I would keep my rating unchanged.
> >
> > [1] Tomczak, J. M., & Welling, M. (2017). VAE with a VampPrior. https://arxiv.org/abs/1705.07120v5
> > [2] Lu, Q., Zhang, Y., Lu, M., & Roychowdhury, V. (2022). Action-conditioned On-demand Motion Generation. Proceedings of the 30th ACM International Conference on Multimedia, 2249–2257. https://doi.org/10.1145/3503161.3548287

---

### Official Review · Reviewer_y3xx · 2023-10-30

**Soundness:** 2 fair
**Presentation:** 3 good
**Contribution:** 2 fair
**Rating:** 3
**Confidence:** 3

**Summary:**

This paper proposes a policy disentangled VAE that can generate diverse videos with user-specified policy during generation. It extends TDVAE by adding policy to the posterior distribution in ELBO. In order to learn the policy from observations, it proposes a lossy compression module and policy module to map observations to a fixed set of latent codes and train it with reconstruction loss.

It experiments in three environments: Moving MNIST, KTH Action and VIZDOOM. The metrics used are video/image quality metrics, e.g. FID, FVD, and policy accuracy. It shows better video quality and policy accuracy compared to MoCoGAN.

**Strengths:**

It studies an interesting problem with a lot of potential applications. For example, it could be used in gaming and movie production and many robotic tasks.

The proposed method PDVAE is consistently better than MoCoGAN in terms of policy accuracy and video quality.

The writing and math in the paper are clear and easy to follow.

**Weaknesses:**

The paper builds on top of TDVAE. Why not compare with TDVAE in terms of video quality regardless that TDVAE cannot generate policy-conditioned rollouts? Can we prompt TDVAE with a demonstration (e.g. digits moving left) to achieve similar results as PDVAE?

The environments and tasks used in the evaluation are very toy cases where the action is always discrete. I wonder if the policy extraction module and the policy-conditioned dynamics model can deal with continuous actions. Does it work with real videos, e.g. driving videos, instead of synthetic ones?

How do you compare with other recent text2video approaches, e.g. Control-A-Video? Will a prompt like “A person running in the style of KTH dataset” already solve the problem?

**Questions:**

How does it compare with TDVAE, which the method is built on?

How does PDVAE work on real videos and continuous action space?

How do you compare with modern text2video diffusion models? Could they solve the task in zero-shot?

---

> ### Author Response · Authors · 2023-11-21
>
> We appreciate your effort and time for the review. Thank you for recognizing the potential applications of our work and the mathematical support of our model. We have written the response for all reviewers for the common concerns. Following is the response to your questions.
>
> 1. How does it compare with TDVAE, which the method is built on?
>
> We have included the additional quantitative evaluations of TD-VAE with the same hyperparameters as PDVAE in Appendix F and below is the table with PDVAE. You can easily think of PDVAE without the policy module and Siam architecture in Figure 3 as the TD-VAE for the comparison.
>
> As you have pointed out, we have thought that TD-VAE is not a valid baseline for the comparison because the contribution of PDVAE is to learn the representation of policy without supervision and the policy-conditioned generative model to control the generation.
>
> 2. How does PDVAE work on real videos and continuous action space?
>
> $\textbf{PDVAE does not explicitly learn the action space as the ELBO in Equation (6) indicates.}$ Our model is agnostic to the action space as it implicitly learns the action space. The common model architecture is used for all three datasets where Moving MNIST and VizDoom have discrete action space, and KTH has the continuous action space. In each transition, the agent moves a certain degree of arms and legs, not a discrete amount. Moreover, KTH is the real videos with low dimensionality. We have elaborated why we have chosen three dataset on the section 3 of common response.
>
> 3. How do you compare with modern text2video diffusion models? Could they solve the task in zero-shot?
>
> We believe text2video diffusion model is not an appropriate comparison to our model. PDVAE and text2video diffusion models can serve similar functionality, but the size of the models and the objective of the model are different. We believe these are two different criteria. It would be great if you provide some more detailed comparisons between models.
>
> 4. About policy conditioned dynamics model
>
> PDVAE is the generative model that implicitly learns the dynamics of the environment for the generation. Our model does not directly learn the dynamics, hence we believe the term dynamics model does not fit. Policy conditioned generative model should be more appropriate.
>
> Thank you.

---

> > ### Comment · Reviewer_y3xx · 2023-11-21
> >
> > Thank you for your response! First, I agree with the other reviewers that the evaluation environments are too simple to prove the superiority of PDVAE. Second, I understand that KTH has continuous action space, but your metric (e.g. pi-ACC) doesn't measure the closeness in continuous space as it relies on discrete labels.

---

> ### Author Response · Authors · 2023-11-22
>
> Dear Reviewer y3xx,
>
> We sincerely appreciate your feedback on our response.
>
> We have wanted to raise the problem of the overlooked intention/behavioral goal behind the action in the field of video generative models. We have hypothesized that the policy from reinforcement learning can be the key to learning such representation. We have derived the ELBO of PDVAE based on TD-VAE. The purpose of the experiments is to validate whether PDVAE can learn the representation and conditionally generate videos by the policy. As you have noticed, PDVAE relies heavily on TD-VAE with the model architecture. We would like to validate our model on high-quality video, but the scalability of TD-VAE (only using simple neural architecture) prohibits us from performing such an experiment. We would appreciate it if you could reconsider our paper with the perspective of learning the representation of policy from unlabeled video and the utilization of policy on the generation.
>
> Secondly, the action space is implicitly modeled with the policy-conditioned generative model. Measuring the performance over the implicit space is hardly possible. Moreover, as we have the action space is not the area of research of this paper, so we have reported the metric (pi-Acc) to evaluate whether PDVAE generates video according to the policy.
>
> Thank you!

---

### Official Review · Reviewer_6csw · 2023-10-31

**Soundness:** 3 good
**Presentation:** 3 good
**Contribution:** 2 fair
**Rating:** 5
**Confidence:** 2

**Summary:**

This paper presents a method (PDVAE) to generate visual representations (video / image sequences) of agents based on policies (and previous states) which are meant to capture the motivation of the agents. This method uses unsupervised learning to learn representations of policies and dynamics of the partially-observable environment without labels. The core ideas are (1) learning a disentangled representation of the policy and state, and (2) adding the policy to the posterior distribution over latent states to derive the ELBO (building on TD-VAE).

**Strengths:**

This paper introduces a novel architecture using VAE that builds on TD-VAE, but separates the representation of the policy from the state. There is a compelling argument that this disentanglement must be done to generate visual representations of agents that exhibit behavior with the appearance of intentionality in dynamic environments.

The paper is not a big improvement over the primary benchmark MoCoGAN, but MoCoGAN has access to the labels that PDVAE does not have. Compared to MoCoGAN- without labels, PDVAE performs much better.

The methodology appears sound, though I have not checked it too closely.

**Weaknesses:**

The writing could be improved, but overall it is fairly clear.

The results overall are not too strong, but they are reasonable and well-motivated, and may be an influence on future work on video generation. Unfortunately, only a single benchmark is used.

Minor Errors:

The authors should add a sentence early on to explain what is meant by "codebook" in the VAE and how this relates to the action space.

Page 4 (Section 3.1): "..given as the observations do..." doesn't quite make sense. Not entirely sure what this is supposed to say.

Page 8 (Section 5.2): "...capable of generating videos generates a video that is not..." --> "...capable of generating videos that are not..."

**Questions:**

Are there no other benchmarks that would be appropriate to compare against other than MoCoGAN?

---

> ### Author Response · Authors · 2023-11-21
>
> We thank you for your time and effort in the review. We appreciate your recognition of our model architecture and argument over the need for policy representation. We have corrected the minor errors that you have indicated on the paper. Please refer to section 2 of the common response for the answer to the question.
>
> Thank you.

---

> > ### Comment · Reviewer_6csw · 2023-11-22
> > **Response to Authors**
> >
> > After considering the reviews of other reviewers and the responses, I have to agree that the experiments are insufficient to make strong claims about the efficacy of the methdology presented.
> >
> > However, I want to reiterate what I wrote in my original review which is that I see value in the approach and it is well-motivated. The core ideas is of general interest. Unfortunately, since the paper does not offer deep theoretical contributions, whether it is ready for publication must be determined by the strength of the experiments and not the general idea.

---

### Author Response · Authors · 2023-11-21

Dear Reviewers,

We sincerely appreciate the time and effort to write meaningful reviews of our work. As the reviewers have similar concerns about our work, we write common responses to all reviewers. We have added more controlled generations for the VizDoom Experiment and comparison with TD-VAE and PDVAE in the supplementary. Moreover, we have added clarification on the contribution to the introduction of the paper. Following are the responses to the concerns in contribution, the single baseline for the evaluation, and the limited evaluation task (absence of high-quality video generation).


1. Contribution

We have raised a problem on the overlooked intention behind the action in the video generative models. There have been plenty of vision models to learn the action or style of the video, but, to the best of our knowledge, the model that learns the representation of intention, behavioral goal, or policy, does not exist.

To learn such representation, we have proposed a novel architecture with ELBO. As mentioned in the last paragraph of the introduction, PDVAE learns and categorizes the policy without label information.  We have validated that PDVAE learns the representation of policy by controlling the generation of the conditioned policy with qualitative and quantitative evaluation.


2. Baseline

To the best of our knowledge, MoCoGAN is the only model that learns the motion-related embedding affecting the entire motion of generated video without supervision.

MoCoGAN has a similar latent variable, called categorical dynamics, to the policy of PDVAE. There are plenty of vision models learning the motion or dynamics-related latent features but these features are responsible for the single transition between frames. MoCoGAN’s categorical dynamics is conditioned to generate the motion features of the entire video, where the agent in the video acts according to the categorical dynamics.

PDVAE unsupervisedly learns the latent variable, policy $\pi$, which accounts for the agent’s strategy to achieve the behavioral goal. During the generation, PDVAE uses the policy $\pi$ and the state $z_t$ to predict the future state $z_{t+1}$, which is decoded to an image $x_{t+1}$ reflecting the policy. Similarly to MoCoGAN, PDVAE uses the policy to generate the entire video, in which the agent acts according to the policy.


3.	Limited evaluation task

The objective of PDVAE is to learn the representation of policy from the unlabeled videos. The model should be able to generate the video conditioned to the policy by reflecting the policy on the agent’s motion and the environment. Moreover, PDVAE should generate diverse videos under the same policy as an agent can act differently under the same policy. We have selected the dataset to validate that PDVAE can learn the policy and generate video conditioned to the policy.

We have chosen $\textbf{Moving MNIST}$ because of its conspicuous result in changing policy. The experiment has been done in TD-VAE from which we have built TD-VAE. We have validated that PDVAE can produce the same result as TD-VAE does with controllability. We have chosen $\textbf{KTH}$ to validate PDVAE works in real-world videos and continuous action space. We have chosen $\textbf{VizDoom}$ to validate PDVAE generates diverse videos in which all agents act differently but according to the conditioned policy.

We believe these three datasets and experimental results are sufficient to validate our contribution. It would be great to validate our model on high-quality video. However, the main objective of PDVAE is not high-quality video generation.

---

### Meta-Review · Area_Chair_JMJX · 2023-11-28

**Metareview:**

**Summary**: This paper uses unsupervised learning to learn representations of policies and dynamics of the partially-observable environment. The proposed method builds on prior work (TD-VAE) by including a policy. Experiments on video datasets show that the method can can generate diverse videos aligned with a specified policy.

**Strengths**:  Reviewers appreciated the clear writing in the paper, the novelty of the method and derivation, and that the proposed method lifted an assumption of prior work (access to labels). The results appeared strong relative to the prior method.

**Weaknesses**: The reviewers' main suggest was to include a more realistic dataset and to compare with additional baselines. The authors contend that some baselines are infeasible to compare against because of computational demands.

There was also a somewhat philosophical point raised in some of the reviews, and again during the discussion: is the proposed method actually inferring _intentions_, or is it just a latent variable model? If the method is ``just'' a latent variable model, then reviewer suggestions to compare with more powerful latent variable models seem valid; if the method is doing something qualitatively different from other latent variable models, then it might be good to showcase those capabilities (and failures of prior methods).

**Justification For Why Not Higher Score:**

Insufficient experimental evaluation. All reviewers voted to reject the paper.

**Justification For Why Not Lower Score:**

N/A

---

### Decision · Program_Chairs · 2024-01-16

Reject